# Predictors of burnout among Belgrade veterinary students: A cross-sectional study

**Jelena Ilić Živojinović**[1], **Dušan Backović**[1], **Goran Belojević**[1], **Olivera Valčić**[2], **Ivan Soldatović**[3], **Janko Janković**[4]*

1 Institute of Hygiene and Medical Ecology, Faculty of Medicine, University of Belgrade, Belgrade, Serbia,
2 Department of Physiology and Biochemistry, Faculty of Veterinary Medicine, University of Belgrade, Belgrade, Serbia, 3 Institute of Medical Statistics and Informatics, Faculty of Medicine, University of Belgrade, Belgrade, Serbia, 4 Institute of Social Medicine, Faculty of Medicine, University of Belgrade, Belgrade, Serbia

* drjankojankovic@yahoo.com

## Abstract

### Background

To the best of our knowledge, studies are lacking on burnout among veterinary students in Serbia, and this is the first study trying to address such a problem. Therefore, the aim of this cross-sectional study was to investigate the predictors of burnout among Belgrade veterinary students.

### Methods

Maslach Burnout Inventory (MBI) and anonymous structured questionnaire addressed to personal data, health habits and stressful influence of educational process were applied among 496 respondents from a total of 1113 students from all grades in spring semester 2014 (response rate 44.6%).

### Results

The prevalence of burnout was 43.3%. High scores on depersonalization and emotional exhaustion scales of MBI were found among 79.4% and 45.0% students, respectively; low personal accomplishment was reported by 50.5% students. Female students reported higher levels of emotional exhaustion compared to males (p = 0.012). A low score on personal achievement scale of MBI was least frequent among the freshmen and most frequent among sophomores (41.1% and 65.3%, respectively; p = 0.986). There were more students with burnout who considered their health as a worsened vs. pre-study period compared to students with no burnout, both concerning mental (61.4% vs. 38.6%) and physical health (61.1% vs. 38.9%; both p<0.001). There were more smokers among students who suffered from burnout compared to students who did not (52.0% vs. 48.0%; p = 0.013). A multiple regression revealed an independent dose-response effect of perceived stress at exams on the onset of burnout (moderate stress OR = 2.164 and high stress OR = 3.878). Also, students with the moderate and high stressful effect of communication with teaching staff, as

**Funding:** The study was funded by the Ministry of Education, Science and Technological Development of the Republic of Serbia (project No.

175078). The funders had no role in study design, data collection and analysis, decision to publish, or preparation of the manuscript.

**Competing interests:** The authors have declared that no competing interests exist.

well as, those with worse self-perceived physical and mental health had more than two times higher presence of burnout.

## Conclusions

The prevalence of burnout among Belgrade veterinary students is relatively high. Primary prevention should be focused on the revealed predictors of burnout.

## Introduction

Numerous negative environmental and psychosocial stressors such as excessive academic obligations, lack of free time, constant pressure to succeed, as well as social determinants of health, including low socioeconomic status, social support, neighborhood environments, exposure to violence, and family conflicts, contribute to emerging health disorders among university students [1–3]. Stress in academic institutions may have a positive effect by increasing self-confidence, but on the other hand, it may negatively affect health in the form of distress [4]. Studying veterinary medicine is often associated with prolonged exposure to academic and non-academic stressors such as: heavy workload [5], efforts to maintain high academic performance [6], difficulty fitting in and unclear expectations [7], hazardous alcohol consumption [8], homesickness and poorer perceived physical health [9] and financial stress [10]. Exposure to work-related stress may continue from academic settings to later veterinarian professional career. Burnout has been increasingly reported in academic and university settings [11–14] and studies from various countries have confirmed that veterinary students have a high risk of burnout [15–17]. The basic characteristics of burnout are feelings of emotional and physical exhaustion, depersonalization, and reduced personal job satisfaction [18].

Concerning health professions, the studies on stress among medical students are dominant, while similar studies among veterinary students are lacking [19,20]. Stress can result in a number of physical and psychological changes which in turn can affect the wellbeing and performance of veterinary students; those who come from calm rural areas to hectic cities are particularly under a raised risk of burnout [21].

There are concerns about the difficulty of a demanding curriculum, time dedicated to learning and the amount of information on all animal species that a veterinary student is expected to adopt [22]. New curricula at the majority of European veterinary schools are adapted to the Bologna process in which greater emphasis is on learning by the student, rather than teaching by the teachers [23]. These curricula introduce a more rigorous program that involves constant checking of students' knowledge through colloquiums and seminars. In addition, the students do not feel academically ready for some of the topics in the first year of studies; it causes loss of interest towards the subjects for which they do not have an adequate knowledge [24,25].

Burnout affects professional development; it may cause reduced professional interest, and further degradation of humanitarian attitudes such as empathy [26]. Monitoring of this syndrome is particularly important at the beginning of studies, when typically, the first symptoms occur. Researches on predictors of burnout are very important for planning and implementation of preventive measures and the use of appropriate coping strategies [27].

In spite of a relatively high risk of burnout among veterinary students, studies of burnout among them are infrequent. For this reason, we have undertaken this very first study on burnout and its correlates among veterinary students in Serbia. We hypothesize that the prevalence of burnout among Belgrade veterinary students is very high.

## Method

### Study design and sample

We performed a cross-sectional study on 496 students (males = 329; 66.3%), of all grades at the Faculty of Veterinary Medicine, University of Belgrade during the spring semester 2014. They study according to a five-year curriculum adapted to the Bologna system. A student takes one colloquium per semester for each subject, as well as a large number of laboratory exercises and seminars. The response rate was 44.6% (496 respondents from 1113 students). The distribution of respondents according to grade was as follows: 124—first year in school, 108—second year in school, 83—third year in school, 66—fourth year in school and 115—fifth year in school. Students were recruited during their laboratory classes and participation was voluntary. A paper questionnaire was completed at the beginning of a class and the instructions were given by the first author. Prior to filling out the questionnaire, all students provided written informed consent to anonymously participate in the study, as well as, gave permission to researchers for using the data.

The study was performed with the permission of the Ethics Committee of the Faculty of Medicine, University of Belgrade (decision No. 29/III-16).

### Study variables

Students completed the Maslach Burnout Inventory (MBI) and an anonymous questionnaire. The Serbian version of MBI is a reliable and valid instrument for measuring burnout, with strong psychometric characteristics of the study instrument, confirmed in a recent Belgrade study [28]. The overall reliability of the scores is sufficient (Cronbach's α = 0.72) with the highest internal consistency value for the scale of emotional exhaustion (Cronbach's α = 0.91) and similar values for depersonalization (DP) and low personal accomplishment (PA). It demonstrates similar reliability to its corresponding original English form, found by Maslach and Jackson [29]. The MBI is a 22-item instrument describing the feelings of a person about his/her job. It consists of three subscales to evaluate each domain of burnout, namely emotional exhaustion (EE), DP and PA [18]. The question "I feel burned out from my work" refers to EE, and the question "I have become more callous toward people since I took this job" is related to DP. "I deal very effectively with the problems of other people" is the question on personal accomplishment [30] Likert scale of seven degrees of the frequency of occurrence of burnout symptoms (0 = never, 1 = a few times a year; 2 = once a month or less; 3 = a few times a month; 4 = once a week; 5 = a few times a week; 6 = every day) is applied in MBI. Scores on each scale can be categorized as low, average or high levels of burnout according to cut-offs detailed in the MBI manual [29]. High EE is defined as scoring ≥27, high DP score is ≥10, and low PA score is ≤33. A person has a burnout if he/she scores high on EE and DP scales [31]. A structured questionnaire included personal data of each student: gender, year of study, number of passed exams, average grade, smoking and drinking habits, physical activity, perceived stressful influence of exams, colloquiums, communication with the teaching staff, contact with the pet owner and field work, self-assessed physical and mental health

### Statistical analysis

The sample size calculation is based on previous studies. It is assumed that the prevalence of burnout in veterinary students is 30%. The sample size of 341 is sufficient to produce a 95% confidence interval with a width of 10% (precision is 5%) when the sample proportion is 30%. In our study sample size reached 496 students in total.

All analyses were performed using SPSS 20 (SPSS Inc., Chicago IL). Descriptive statistics included count (percent) or mean (standard deviation) depending on the data type. Groups were compared using Pearson chi-square or Cochran-Armitage test for trend for attributive data (nominal and ordinal) and t-test and ANOVA for numerical data. Numerical data were examined for normal distribution. Normal distribution was evaluated using graphical methods (Q-Q plot, histogram, boxplot), descriptive statistics (mean, standard deviation and median) and tests for normality (Kolmogorov-Smirnov).

Multiple regression analysis was used to evaluate the investigated factors that independently correlate with burnout (gender, self-assessed physical and mental health and stressful influence of exams, communication with the teaching staff and field work). The dependent variable was a high combined score on subscales EE and DP of MBI [31]. Independent variables included significant predictors in univariate analysis. Modeling was performed in several steps. First, the enter method [32] was employed. The probability of alpha error less than 0.05 was considered significant.

## Results

The majority of students reported high levels of DP (79.4%); about half of them experienced low levels of PA or high levels of EE (50.5% and 45%, respectively) (Table 1). Overall, 43.3% of participants met the criteria for burnout.

There was no significant impact of gender or year of study on the prevalence of burnout. Females more frequently reported higher levels of EE compared to male students. A low PA score was least frequent among freshmen (41.1%) and most frequent among the second year (65.3%) and fourth-year students (50.8%).

There was no statistically significant difference neither in mean grade (p = 0.811) nor in the number of passed exams (p = 0.809), physical activity (p = 0.785) and alcohol consumption (0.104) between the students with and without burnout (Table 2).

There were more smokers among students who suffered from burnout (52.0%) compared to students who did not (48.0%, p = 0.013). The majority of students with burnout considered their mental and physical health worsened compared to the pre-study period (p = 0.001) (Table 2).

Student's responses to the perceived stress impact of exams resulted in a highly significant difference in relation to the presence of burnout (p<0.001) (Table 3).

The highest percentage of students who perceived exams as very stressful was in the group with burnout (58.8%). Also, students with burnout perceived colloquiums, communications with teaching staff and contact with pet owners as more stressful in comparison to students without burnout (Table 3).

Missing value analysis is performed to evaluate missing values in the database. Majority of variables have percent of missing values less than 1%, but variables Passed exams has 24 missings (4.8%), Mean grade 32 (6.5%), MBIPAsum as well as MBIPAlow 39 (7.9%), Physical activity 23 (4.6%), Field work 42 (8.5%) and Contact with animal owners 25 (5.0%). These missing values are assumed to be missing at random.

Multiple regression analysis revealed an independent dose-response effect of perceived stress at exams on the onset of burnout (moderate stress OR = 2.164 and high stress OR = 3.878). Also, students with the moderate and high stressful effects of communication with teaching staff, as well as, those with worse self-perceived physical and mental health had more than two times higher presence of burnout (Table 4).

The model reveals that all predictors except gender are significant to predict burnout outcome. Health worsening (physical and mental) significantly favors burnout outcome. Stress-

**Table 1. MBI subscale scores and burnout prevalence by student gender and year of study.**

| Variables | Emotional exhaustion (EE) | | Depersonalization (DP) | | Personal accomplishment (PA) | | Burnout n (%) |
|---|---|---|---|---|---|---|---|
| | Mean±SD | high EE score n (%) | Mean ±SD | high DP score n (%) | Mean ±SD | low PA score n (%) | |
| All (n = 496) | 25.60 ±11.61 | 223 (45.0) | 15.40 ±6.56 | 394 (79.4) | 32.18 ±8.50 | 271 (50.5) | 215 (43.3) |
| Gender | | | | | | | |
| Male (n = 329) | 24.67 ±11.74 | 141 (42.9) | 15.33 ±6.57 | 262 (79.6) | 32.65 ±8.74 | 151 (49.0) | 136 (41.3) |
| Female (n = 167) | 27.44 ±11.16 | 82 (49.1) | 15.56 ±6.56 | 132 (79.0) | 31.21 ±7.94 | 80 (53.7) | 79 (47.3) |
| p value | 0.012[a] | 0.186[b] | 0.710[a] | 0.877[b] | 0.090[a] | 0.350[b] | 0.205[b] |
| Year of study (n) | | | | | | | |
| I (124) | 24.33 ±11.37 | 50 (40.3) | 15.64 ±6.44 | 104 (83.9) | 33.70 ±8.01 | 46 (41.1) | 50 (40.3) |
| II (108) | 26.12 ±11.24 | 52 (48.1) | 15.81 ±6.69 | 84 (77.8) | 29.58 ±9.14 | 66 (65.3) | 51 (47.2) |
| III (83) | 27.53 ±11.819 | 41 (49.4) | 15.73 ±7.10 | 67 (80.7) | 32.21 ±9.09 | 37 (48.7) | 38 (45.8) |
| IV (66) | 24.80 ±11.96 | 29 (43.9) | 14.09 ±6.37 | 47 (71.2) | 32.97 ±7.26 | 31 (50.8) | 26 (39.4) |
| V (115) | 25.57 ±11.84 | 51 (44.3) | 15.29 ±6.27 | 92 (80.0) | 32.59 ±8.21 | 51 (47.7) | 50 (43.5) |
| p value | 0.367[c] | 0.720[d] | 0.486[c] | 0.325[d] | 0.008[c] | 0.986[d] | 0.952[d] |

SD standard deviation; Burnout if high in EE $\geq$27 and DP $\geq$10

Post Hoc Test–mean grades: I vs. III p = 0.034; II vs. IV p = 0.007; III vs. IV p = 0.002; IV vs. V p = 0.029

[a]Independent Samples T test

[b]Pearson Chi-Square

[c]ANOVA

[d]Chi-Square test for trend (Cochran Armitage test)

related to examination and communication with teaching stuff category gradation correlates with higher odds for burnout outcome. Stressful effect of fieldwork has a nearly significant relationship (p = 0.062), but only category "mild" vs. no stress at all.

The presented model has explained variability 28.1% (Nagelkerke R square = 0.281) and the model is calibrated (Hosmer and Lemeshow test p = 0.878). Based on the area under the curve, the model has good discriminative power (c = 0.77195%; CI 0.728–0.815; p<0.001).

## Discussion

This study investigated the prevalence of burnout and predictors contributing to its experience among Belgrade veterinary students. Our results indicated that 43.3% of participants met the criteria for burnout; this prevalence is higher compared to a recent study among Australian veterinary students [16] where a high risk of burnout was found among 30% of them. A study among UK veterinary students [17] showed that 58% of the respondents reported low self-esteem and/or depression, which may be the predictors for burnout. They also had poorer wellbeing and a higher degree of mental distress in comparison to the general population [17]. The survey conducted among 289 students of the College of Veterinary Medicine in Tennessee, showed much more common symptoms of depression and stress than in the general

**Table 2. Academic achievement, health behavior and self-assessed health status with regard to burnout among Belgrade veterinary students.**

| Variables | Burnout | | p value |
|---|---|---|---|
| | **No** | **Yes** | |
| Passed exams | 21.88±14.21 | 22.17±14.01 | 0.809[a] |
| Mean grade | 8.07±0.66 | 8.06±0.74 | 0.811[b] |
| Mental health compared to pre-study period, n (%) | | | <0.001[c] |
| No change | 150 (73.9) | 53 (26.1) | |
| Better | 38 (69.1) | 17 (30.9) | |
| Worse | 91 (38.6) | 145 (61.4) | |
| Physical health compared to pre-study period, n (%) | | | <0.001[c] |
| No change | 172 (69.4) | 76 (30.6) | |
| Better | 22 (75.9) | 7 (24.1) | |
| Worse | 84 (38.9) | 132 (61.1) | |
| Smokers, n (%) | | | 0.013[c] |
| No | 206 (60.1) | 137 (39.9) | |
| Yes | 71 (48.0) | 77 (52.0) | |
| Physical activity, n (%) | | | 0.785[c] |
| No | 119 (56.9) | 90 (43.1) | |
| Yes | 147 (55.7) | 117 (44.3) | |
| Alcohol consumption, n (%) | | | 0.104[c] |
| No | 141 (60.5) | 92 (39.5) | |
| Yes | 139 (53.3) | 122 (46.7) | |

[a]Mann-Whitney U test

[b]T test

[c]$\chi$2test

population [21]. Another US study conducted among first-year veterinary students of Kansas State University College, reported high levels of depression and anxiety, with significant predictors such as poor physical health, difficulty fitting in among colleagues and high academic expectations [33]. These findings are congruent with our results showing that burnout and poor subjectively estimated physical health may be expected in every second veterinary student.

Many studies have confirmed that females are more susceptible to stress and burnout compared to males [21,34,35]. The results of our study showed that there is no significant impact of gender on the prevalence of burnout, although females more frequently reported higher levels of EE compared to male students. The study conducted in California found that living arrangements, specifically whether a student lived on his/her own was associated with burnout, which suggests that veterinary students may benefit from being a part of a support system whether in a community or at the university [15]. Perhaps the causes of stress should be sought more in the course of their studies. This means that students should better connect with colleagues, mentors and support staff to overcome homesickness. By sponsoring sports events aimed at improving physical health, it would also have a positive effect on students' mental functioning. By providing adequate resources for problem-solving and developing collective empathy, the prejudices associated with seeking help for mental health would be overcome [33].

In our study, the highest mean value of the EE scale was observed in the third year of study when the curriculum is the most comprehensive and when clinical subjects begin. In most

**Table 3. Perceived stressful influence of study activities with regard to burnout among Belgrade veterinary students.**

| Perceived stress | Burnout (No) | Burnout (Yes) | p value[a] |
|---|---|---|---|
| Exams, n (%) | | | <0.001 |
| Absent | 17 (85.0) | 3 (15.0) | |
| Mild | 45 (84.9) | 8 (15.1) | |
| Moderate | 123 (63.7) | 70 (36.3) | |
| High | 93 (41.2) | 133 (58.8) | |
| Colloquiums, n (%) | | | <0.001 |
| Absent | 78 (78.8) | 21 (21.2) | |
| Mild | 112 (58.6) | 79 (41.4) | |
| Moderate | 70 (43.5) | 91 (56.5) | |
| High | 17 (44.7) | 21 (55.3) | |
| Communication with the teaching staff, n (%) | | | <0.001 |
| Absent | 141 (73.1) | 52 (26.9) | |
| Mild | 89 (53.3) | 78 (46.7) | |
| Moderate | 33 (37.19) | 56 (62.9) | |
| High | 14 (33.3) | 28 (66.7) | |
| Field work, n (%) | | | <0.001 |
| Absent | 180 (62.3) | 109 (37.7) | |
| Mild | 56 (48.3) | 60 (51.7) | |
| Moderate | 13 (48.1) | 14 (51.9) | |
| High | 9 (40.9) | 13 (59.1) | |
| Contact with pet owners, n (%) | | | <0.001 |
| Absent | 186 (63.1) | 109 (36.9) | |
| Mild | 55 (50.5) | 54 (49.5) | |
| Moderate | 21 (39.6) | 32 (60.4) | |
| High | 5 (35.7) | 9 (64.3) | |

[a]Trend = Mantel–Haenszel chi square test for trend

veterinary programs, students of the first year do not start clinical rotations and are not exposed to stressors related to clinical work, such as dissatisfied clients, death of pets or diagnostic errors [36]. Adequate mechanisms are needed to help veterinary students to cope with the stress that is more intense and cumulative in higher studies, especially in the third year, which is most stressful given the greater clinical responsibility that students assume, the possible subsequent diagnostic errors, and the death of the patients they encounter [36]. Veterinary students experience psychological and physiological changes during their education due to perceived poor physical health, unclear expectations, difficulties in fitting in and heavy workload. The highest anxiety and depression among US students were found in their second and third year of studying [7]. In our research, the percentage of students with burnout was the highest among sophomores.

Our result of poorer self-estimated physical and mental health in the group of students with burnout compared to the group without burnout has been confirmed as well as in the other studies like a predictor of anxiety and depression in veterinary students [7,33].

Health-related habits such as exercise could play important roles in the mediation between psychological distress and coping styles influencing mental health [37]. Reduced physical activity could be associated with stress among university students [38]. Our results showed no statistically significant differences in physical activity between the students with burnout and without it. Maybe the explanation is in the extremely heavy workload which does not leave

**Table 4. Multiple regression model with the presence of burnout as a dependent variable and perceived stress of academic activities as independent factors.**

| Variables | OR (95% CI) | p value |
|---|---|---|
| Stressful effect of exam | | |
| No | 1 | 0.002 |
| Mild | 0.894 (0.182–4.400) | 0.890 |
| Moderate | 2.164 (0.527–8.884) | 0.284 |
| High | 3.878 (0.944–15.923) | 0.060 |
| Stressful effect of communication with teaching staff | | |
| No | 1 | 0.014 |
| Mild | 1.724 (1.028–2.892) | 0.039 |
| Moderate | 2.531 (1.339–4.785) | 0.004 |
| High | 2.690 (1.126–6.427) | 0.026 |
| Stressful effect of field work | | |
| No | 1 | 0.208 |
| Mild | 1.638 (0.976–2.751) | 0.062 |
| Moderate | 1.875 (0.732–4.804) | 0.190 |
| High | 1.045 (0.375–2.914) | 0.932 |
| Estimated physical health | | |
| No change | 1 | 0.003 |
| Better | 0.675 (0.252–1.809) | 0.434 |
| Worse | 2.110 (1.324–3.364) | 0.002 |
| Estimated mental health | | |
| No change | 1 | 0.005 |
| Better | 1.507 (0.688–3.304) | 0.306 |
| Worse | 2.307 (1.390–3.830) | 0.001 |
| Gender (female) | 0.708 (0.444–1.129) | 0.147 |

students enough free time for physical activity beside rest and sleep [39]. In accordance with the latest recommendations of the Center for Disease Control on the benefits of physical activity for psychological health, it is certainly necessary to encourage students to exercise on a regular basis in order to overcome stress and improve their health [39].

Regarding other aspects of health habits, smoking was more frequent among students with burnout compared to students without it (p = 0.013). Our results on healthy habits support the findings of similar studies indicating the connection between smoking and alcohol consumption with stress [31], but more detailed research to determine this connection is needed. The explanation of this link could be found in the fact that healthcare practitioners are often exposed to a burden of duties and responsibilities at work and use tobacco and alcohol as legal ways of relaxing [40].

There are evidence that major predictors of academic burnout are: the absence of free time, fear of failure, lack of help and support, uncertain future, big pressure due to exams, bad financial situation as well as stressful contacts with patients [41]. Research among medical students showed that stress related to exams, but not to colloquiums and other teaching activities was an important predictor of burnout [42–44]. In our study, both exams, colloquiums, communications with teaching staff and contacts with pet owners were stressful for veterinary students. Finally, the perceived stress at the exams remained as an independent correlate of students' burnout in a multivariate analysis.

The study conducted in Australia determined that students of veterinary medicine also seem to suffer from high levels of anxiety and stress and have inadequate strategies for coping

with adversity [25]. Modifying the curriculum at the Faculty of Veterinary Medicine, which includes lectures on skills for overcoming stress and stressful situations, may enable future veterinarians to improve their skills in the workplace and their mental health, which would contribute to a higher satisfaction at their workplace. The integration of communication and coping skills and leadership, within education in the curricula of veterinary schools may improve the present situation [45,46]. On-line learning resources [47] together with online tutoring support [48] can incorporate student-centered learning in basic subjects which can help students to overcome stress. Communication skills can also be effectively taught in experimental learning, discussions, and feedback through role-play in small groups of students using simulated clients and trained facilitators [49].

To the best of our knowledge, this is the first study on burnout and its correlates among veterinary students in Serbia. A variety of different academic and non-academic stressors, such as heavy workload, fitting in with peers, rigorous academic requirements, home sickness, relationship difficulties, and financial strains are common to university students. Thus, the results of our research may be generalized to veterinary students worldwide. However, several limitations should be mentioned. First, the use of cross-sectional design does not allow a causal relationship to be determined among variables. Second, the information on all variables had been self-reported and may have been subject to recall bias. Third, a limitation of the study may be the selection bias because students affected by burnout may be more willing to participate as suggested by other burnout scales not used in our study. Fourth, burnout in the workplace may be also related to administrative policy/procedure that was not measured in our study. Fifth, Serbian students and their families due to a turbulent recent history of heavy economic crisis and war may have improved their resilience to stress compared to the veterinary students from other European regions. Furthermore, resilience, grit and other measures of improved mental wellbeing were not explored in this study, nor the concept of comparing highly and poorly resilient people and their predisposition (or not) to burnout.

## Conclusion

Notwithstanding all previously mentioned limitations, our study shows that the prevalence of burnout among Belgrade veterinary students is relatively high. In univariate analysis the main correlates of students' burnout are female gender, smoking, poor subjectively estimated physical and mental health and perceived stress from academic activities. In a multiple regression analysis, the independent dose-response effect on the onset of burnout remained only for perceived stress at the exams.

The study results indicate that it is possible to determine proneness to burnout among veterinary students. There are some future considerations concerning curriculum adjustment or a different time planning of colloquiums that can leave more time to relax and lower the risk of burnout. The ntroduction of peer support and culture of acceptance and hospitality may be helpful for students' better adjustment.

## Author Contributions

**Conceptualization:** Jelena Ilić Živojinović, Dušan Backović.

**Data curation:** Jelena Ilić Živojinović, Olivera Valčić, Janko Janković.

**Formal analysis:** Ivan Soldatović.

**Funding acquisition:** Goran Belojević.

**Methodology:** Jelena Ilić Živojinović, Dušan Backović.

**Supervision:** Dušan Backović, Goran Belojević.

**Writing – original draft:** Jelena Ilić Živojinović, Dušan Backović.

**Writing – review & editing:** Goran Belojević, Olivera Valčić, Ivan Soldatović, Janko Janković.

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
