## [Decision Letter · Decision Letter 0]

30 Jan 2020

PONE-D-19-32894

Predictors of burnout among Belgrade veterinary students: a cross-sectional study

PLOS ONE

Dear Dr. Janković,

Thank you for submitting your manuscript to PLOS ONE. After careful consideration, we feel that it has merit but does not fully meet PLOS ONE’s publication criteria as it currently stands. Therefore, we invite you to submit a revised version of the manuscript that addresses the points raised during the review process.

Two reviewers have assessed your work. Although one of them advises against the consideration of your paper, I would like to say that after a careful review of the manuscript, I believe (also based in the comments provided by the second reviewer) that the work has a certain potential and could be revised by the authors. Thus, based in the comments/suggestions provided by this reviewer, please consider performing an extensive revision of the paper. Afterwards, I will submit it for and a second evaluation from two academic reviewers.

We would appreciate receiving your revised manuscript by Mar 15 2020 11:59PM. To enhance the reproducibility of your results, we recommend that if applicable you deposit your laboratory protocols in protocols.io, where a protocol can be assigned its own identifier (DOI) such that it can be cited independently in the future. For instructions see: http://journals.plos.org/plosone/s/submission-guidelines#loc-laboratory-protocols

We look forward to receiving your revised manuscript.

Kind regards,

Sergio A. Useche, Ph.D.

Academic Editor

PLOS ONE

Journal Requirements:

2. Please provide additional details regarding participant consent.

In the Methods section, please ensure that you have specified what type of consent you obtained (for instance, written or verbal) and whether the ethics committee approved this consent procedure.

If verbal consent was obtained please state why it was not possible to obtain written consent and how verbal consent was recorded. If your study included minors, state whether you obtained consent from parents or guardians.

Reviewers' comments:

Reviewer's Responses to Questions

**Comments to the Author**

1. Is the manuscript technically sound, and do the data support the conclusions?

Reviewer #1: Yes

Reviewer #2: Partly

2. Has the statistical analysis been performed appropriately and rigorously? 

Reviewer #1: N/A

Reviewer #2: No

3. Have the authors made all data underlying the findings in their manuscript fully available?

Reviewer #1: Yes

Reviewer #2: Yes

4. Is the manuscript presented in an intelligible fashion and written in standard English?

Reviewer #1: Yes

Reviewer #2: Yes

5. Review Comments to the Author

Reviewer #1: this is not an appropriate journal for this type of study. A veterinary or mainstream education journal would be more appropriate. It seems silly for me to keep writing to ensure that minimum characters are met before I can submit this review!!!! PLoS One needs a better system to avoid discouraging reviewers.

Reviewer #2: Thank you for the opportunity to review your manuscript. It is an important addition to the veterinary and mental health literature. Please consider these comments for improvement and clarity.

Line 47: "walkability" is not a commonly used English word and does not transmit a known stressor.

Line 49: "eustress" is spot-on but uncommonly used. You may consider defining it or selecting a more common word to indicate your intent.

Line 78: This is the Knowledge Gap and can be more accurate if "still lacking" was changed to "infrequent".

Line 80: This is the Hypothesis and is somewhat vague due to the suggestion that "prevalence...is significant". The remainder of that sentence is not needed as you are not testing interventions.

Line 90 and 91: perhaps changing the word "grade" to "year in school" may be more appropriate.

Line 101: The Belgrade study should be referenced. Is this #28?

Line 105: Should ref #28 here be ref #30? If so, some rearrangement of reference order is needed.

Line 108: Should ref #13 here be ref #18?

Line 121L Statistical analysis: The deficiencies here should be addressed to better understand and clarify your approach. A power calculation was not completed or reported and tests for normality were not mentioned. You have chosen parametric statistics but never indicated proof that your data was normally distributed.

Line 137: Table 1: There are some inconsistencies which make interpretation very questionable. For instance: the number of male (325) and female (165) participants do not equal the total of all genders (496). Also similarly, the number of male (134) and female (79) participants do not equal the total of all genders (215) with burnout. Many of the N and percentages do not match, therefore, please review and revise all the data in this Table. Without that detail, there is too much inconsistency to believe your data and conclusions.

Line 139:Improvement to remind the reader that >= 27 for EE and >=10 for DP are your cut-offs for burnout. "High" is relative and the level at which you analyzed your data and made your interpretations must be crystal clear.

Line 141: "Cocarame" should be "Cochran"

Line 144: Again, relative relationships must be avoided in Results and are better left to the Conclusions. Please provide specific data instead of "low" PA score. This should be the pattern throughout.

Line 154: Please insert a comma between the parameter and "n (%)" throughout this table. Ensure that the "n" is always present. Also in Table 2 and Table 3, the p value is not always clearly indicating which parameters it is comparing. Do you only provide the statistically significant data for the parameter that meets significance? Sometimes the p value appears to be on a specific line and other times it appears to split the line spacing.

Line 177&178: This seems to be a nonsequitor. Perhaps it is the right sentiment but needs more explanation to clarify to the reader.

Line 187: This study from Tennessee ("College" of Veterinary Medicine, not "Faculty") had 289 students not 389 as written. Also, there are contradictory versions of this study depending on the link followed from pubmed - one abstract is the Tennessee description. Another one follows a report from Oregon State University. Please be sure of your reference and trail of information.

Line 189: "Faculty" should read "Kansas State Univeristy College"

Line 200: Ref 33 may suggest investigation into the causes of stress in a students studies or implementation of programs to reduce the stressors is needed - which is it you seek in this line? Please clarify and allow for completion of your thought in this paragraph.

Line 207: This is not a study of veterinary students, please amend as it implies that this is known in DVMs but it is not.

Line 224: The comment "(borderline significance)" is incorrect. Data is either significant upon your predetermined criteria and level or it is not. Amend this, please.

Line 251: The limitations can be improved overall. There are many limitations that were not nor are not measured in your study that may contribute to burnout. You may mention other burnout scales or references that suggest those with burnout are more apt to complete the survey. The mention of the recent war in Serbia could also be viewed to improve resilience and not to lowered their ability to cope. You do not mention resilience, grit, or other measures of improved mental wellbeing anywhere in your manuscript. The limitations may be a good place to introduce that concept as you were not, at the outset, looking to compare highly and poorly resilient people and their predisposition (or not) to burnout. Also recent publications that discuss burnout in the workplace that may be most related to administrative policy/procedure and not work load could be mentioned.

6. PLOS authors have the option to publish the peer review history of their article (what does this mean?). If published, this will include your full peer review and any attached files.

Reviewer #1: No

Reviewer #2: Yes: Karl E Jandrey

---

## [Author Response · Author response to Decision Letter 0]

23 Feb 2020

Response to the Reviewer’s Comments

Reviewer #2: Thank you for the opportunity to review your manuscript. It is an important addition to the veterinary and mental health literature. Please consider these comments for improvement and clarity.

We want to extend our appreciation for reviewer#2 taking the time and effort necessary to provide such insightful guidance. We carefully considered your comments and hope that these revisions improve the manuscript such that you now deem it worthy of publication in PLOS ONE. All our responses are highlighted in yellow.

COMMENTS

Reviewer: Line 47, "walkability" is not a commonly used English word and does not transmit a known stressor.

RESPONSE: The word "walkability" was deleted.

Reviewer: Line 49, "eustress" is spot-on but uncommonly used. You may consider defining it or selecting a more common word to indicate your intent.

RESPONSE: The word "eustress" was replaced with “Stress in academic institutions may have a positive effect by increasing self-confidence “. 

Reviewer: Line 78, This is the Knowledge Gap and can be more accurate if "still lacking" was changed to "infrequent".

RESPONSE: The phrase "still lacking" was replaced with “infrequent “.

Reviewer: Line 80, This is the Hypothesis and is somewhat vague due to the suggestion that "prevalence...is significant". The remainder of that sentence is not needed as you are not testing interventions.

RESPONSE: The sentence "prevalence...is significant" was changed into "prevalence...is very high" and the remainder of that sentence is deleted.

Reviewer: Line 90 and 91, perhaps changing the word "grade" to "year in school" may be more appropriate.

RESPONSE: The word "grade" was changed in "year in school" which is more appropriate.

Reviewer: Line 101, The Belgrade study should be referenced. Is this #28?

RESPONSE: The Belgrade study is referenced as 28.

Reviewer: Line 105, Should ref #28 here be ref #30? If so, some rearrangement of reference order is needed.

RESPONSE: The ref #28 is changed into ref #29 (ex ref#30) and we made rearrangement of the reference order. 

Reviewer: Line 108, Should ref #13 here be ref #18?

RESPONSE: The ref #13 is #18, as we changed.

Reviewer: Line 121, Statistical analysis: The deficiencies here should be addressed to better understand and clarify your approach. A power calculation was not completed or reported and tests for normality were not mentioned. You have chosen parametric statistics but never indicated proof that your data was normally distributed.

RESPONSE: In methods section, Statistical analysis, explanation regarding normality of data is added to clarify chosen tests. Sample size calculation is, also, added into statistical methodology.

The following is added: “Sample size calculation is based on previous studies. It is assumed that prevalence of burnout in veterinary students is 30%. Sample size of 341 is sufficient to produce 95% confidence interval with a width of 10% (precision is 5%) when the sample proportion is 30%. In our study sample size reached 496 students in total. Numerical data were examined for normal distribution. Normal distribution was evaluated using graphical methods (Q-Q plot, histogram, boxplot), descriptive statistics (mean, standard deviation and median) and tests for normality (Kolmogorov-Smirnov). “

Reviewer: Line 137, Table 1: There are some inconsistencies which make interpretation very questionable. For instance: the number of male (325) and female (165) participants do not equal the total of all genders (496). Also, similarly, the number of male (134) and female (79) participants do not equal the total of all genders (215) with burnout. Many of the N and percentages do not match, therefore, please review and revise all the data in this Table. Without that detail, there is too much inconsistency to believe your data and conclusions.

RESPONSE: The database was revised and several inconsistencies about data were found (those participants with missing values on Maslach Burnout Inventory were extracted from the database). Consequently, statistical analysis was performed again and changes were made regarding data in all tables. In Results section, new tables are presented instead of the old ones. No significant changes have been observed in new tables, but these tables are consistent and more accurate compared to the old ones. 

Still, missing values exists in database as expected in anonymous questionnaires. 

Missing value analysis is performed to evaluate missing values in database. Majority of variables have percent of missing values less than 1%, but variables Passed exams has 24 missing (4.8%), Mean grade 32 (6.5%), MBIPAsum as well as MBIPA.low 39 (7.9%), Physical activity 23 (4.6%), Field work 42 (8.5%) and Contact with animal owners 25 (5.0%). These missing values are assumed to be missing at random. The missing value analysis report is added to Results section.

Reviewer: Line 139, Improvement to remind the reader that >= 27 for EE and >=10 for DP are your cut-offs for burnout. "High" is relative and the level at which you analyzed your data and made your interpretations must be crystal clear.

RESPONSE: It was added as suggested to remind the reader that >= 27 for EE and >=10 for DP are cut-offs for burnout. 

Reviewer: Line 141, "Cocarame" should be "Cochran"

RESPONSE: We changed "Cocarame" in "Cochran".

Reviewer: Line 144, Again, relative relationships must be avoided in Results and are better left to the Conclusions. Please provide specific data instead of "low" PA score. This should be the pattern throughout.

RESPONSE: We have provided specific data for the required parameters throughout the Results section. 

"A low PA score was least frequent among freshmen (41.1%) and most frequent among second year (65.3%) and fourth year students (50.8%)….."

Reviewer: Line 154, Please insert a comma between the parameter and "n (%)" throughout this table. Ensure that the "n" is always present. Also in Table 2 and Table 3, the p value is not always clearly indicating which parameters it is comparing. Do you only provide the statistically significant data for the parameter that meets significance? Sometimes the p value appears to be on a specific line and other times it appears to split the line spacing.

RESPONSE: A comma is inserted between the parameter and n (%) in Table 2 and "n" is always present. We added new tables instead of the old ones. We have arranged Table 2 and Table 3 so that the p value clearly indicates which parameter it is comparing. We provided the statistically significant data for all parameters, not only for those that meet significance.

Reviewer: Line 177 & 178, This seems to be a nonsequitor. Perhaps it is the right sentiment but needs more explanation to clarify to the reader.

RESPONSE: Explanation was added.

Reviewer: Line 187, This study from Tennessee ("College" of Veterinary Medicine, not "Faculty") had 289 students not 389 as written. Also, there are contradictory versions of this study depending on the link followed from pubmed - one abstract is the Tennessee description. Another one follows a report from Oregon State University. Please be sure of your reference and trail of information.

RESPONSE: Thanks for your observation. All suggested changes were made and the reference was checked.

Reviewer: Line 189, Faculty" should read "Kansas State Univeristy College"

RESPONSE: We changed "Faculty" with "Kansas State University College".

Reviewer: Line 200, Ref 33 may suggest investigation into the causes of stress in a students studies or implementation of programs to reduce the stressors is needed - which is it you seek in this line? Please clarify and allow for completion of your thought in this paragraph.

RESPONSE: We clarified ref 33. “This means that students should better connect with colleagues, mentors and support staff to overcome homesickness. By sponsoring sports events aimed at improving physical health, it would also have a positive effect on students' mental functioning. By providing adequate resources for problem solving and developing collective empathy, the prejudices associated with seeking help for mental health would be overcome “. 

Reviewer: Line 207, This is not a study of veterinary students, please amend as it implies that this is known in DVMs but it is not.

RESPONSE: The text refers to ref 36, not 37, and it was revised.

Reviewer: Line 224, The comment "(borderline significance)" is incorrect. Data is either significant upon your predetermined criteria and level or it is not. Amend this, please.

RESPONSE: We added the p value for smoking habits and deleted "(borderline significance)". It seems that alcohol consumption was not significant (p=0.104) in terms of new statistical analyses so we deleted it also. 

"Regarding other aspects of health habits, smoking was more frequent among students with burnout compared to students without it (p = 0.013). " 

Reviewer: Line 251, The limitations can be improved overall. There are many limitations that were not nor are not measured in your study that may contribute to burnout. You may mention other burnout scales or references that suggest those with burnout are more apt to complete the survey. The mention of the recent war in Serbia could also be viewed to improve resilience and not to lowered their ability to cope. You do not mention resilience, grit, or other measures of improved mental wellbeing anywhere in your manuscript. The limitations may be a good place to introduce that concept as you were not, at the outset, looking to compare highly and poorly resilient people and their predisposition (or not) to burnout. Also recent publications that discuss burnout in the workplace that may be most related to administrative policy/procedure and not work load could be mentioned.

RESPONSE: The limitations were revised and improved.

---

## [Decision Letter · Decision Letter 1]

6 Mar 2020

Predictors of burnout among Belgrade veterinary students: a cross-sectional study

PONE-D-19-32894R1

Dear Dr. Janković,

We are pleased to inform you that your manuscript has been judged scientifically suitable for publication and will be formally accepted for publication once it complies with all outstanding technical requirements.

With kind regards,

Sergio A. Useche, Ph.D.

Academic Editor

PLOS ONE

Additional Editor Comments (optional):

Reviewers' comments:

Reviewer's Responses to Questions

**Comments to the Author**

1. If the authors have adequately addressed your comments raised in a previous round of review and you feel that this manuscript is now acceptable for publication, you may indicate that here to bypass the “Comments to the Author” section, enter your conflict of interest statement in the “Confidential to Editor” section, and submit your "Accept" recommendation.

Reviewer #2: All comments have been addressed

2. Is the manuscript technically sound, and do the data support the conclusions?

Reviewer #2: Yes

3. Has the statistical analysis been performed appropriately and rigorously? 

Reviewer #2: Yes

4. Have the authors made all data underlying the findings in their manuscript fully available?

Reviewer #2: No

5. Is the manuscript presented in an intelligible fashion and written in standard English?

Reviewer #2: Yes

6. Review Comments to the Author

Reviewer #2: Thank you for carefully re-evaluating your data and updating the manuscript in a more clear and understandable condition. There are aa few very minor language irregularities in some of the newer text which the editor will likely catch and amend. The concept and information you have presented is in a publishable stage.

7. PLOS authors have the option to publish the peer review history of their article (what does this mean?). If published, this will include your full peer review and any attached files.

Reviewer #2: Yes: Karl Jandrey, DVM, MAS, DACVECC

---

## [Editor Report · Acceptance letter]

11 Mar 2020

PONE-D-19-32894R1 

Predictors of burnout among Belgrade veterinary students: a cross-sectional study 

Dear Dr. Janković:

I am pleased to inform you that your manuscript has been deemed suitable for publication in PLOS ONE. Congratulations! Your manuscript is now with our production department. 

With kind regards,

on behalf of

Dr. Sergio A. Useche 

Academic Editor

PLOS ONE